# A Highly Sensitive 3D Resonator Sensor for Fluid Measurement

**DOI:** 10.3390/s23146453

**Published:** 2023-07-17

**Authors:** Ali M. Almuhlafi, Omar M. Ramahi

**Affiliations:** 1Electrical Engineering Department, King Saud University, Riyadh 11421, Saudi Arabia; 2Department of Electrical and Computer Engineering, University of Waterloo, Waterloo, ON N2L 3G1, Canada; oramahi@uwaterloo.ca

**Keywords:** microwaves, split-ring resonators, complementary split-ring resonators, complementary electric-LC resonator, fluid-level detection, sensitivity enhancement

## Abstract

Planar sub-wavelength resonators have been used for sensing applications, but different types of resonators have different advantages and disadvantages. The split ring resonator (SRR) has a smaller sensing region and is suitable for microfluidic applications, but the sensitivity can be limited. Meanwhile, the complementary electric-LC resonator (CELCR) has a larger sensing region and higher sensitivity, but the topology cannot be easily designed to reduce the sensing region. In this work, we propose a new design that combines the advantages of both SRR and CELCR by incorporating metallic bars in a trapezoid-shaped resonator (TSR). The trapezoid shape allows for the sensing region to be reduced, while the metallic bars enhance the electric field in the sensing region, resulting in higher sensitivity. Numerical simulations were used to design and evaluate the sensor. For validation, the sensor was fabricated using PCB technology with aluminum bars and tested on dielectric fluids. The results showed that the proposed sensor provides appreciably enhanced sensitivity in comparison to earlier sensors.

## 1. Introduction

Nowadays, sensitive and affordable measurement systems are important for a wide range of applications such as lab-on-a-chip, point-of-care lab testing, rapid testing for quality control, environmental monitoring, and public health and safety [1]. These applications require highly sensitive and inexpensive measurement systems. Planar sub-wavelength resonators have emerged as strong candidates for designing sensitive near-field sensors due to their ability to concentrate electromagnetic fields in a small volume surrounding the resonator [2]. By perturbing these fields through topological or material changes, the impedance of the sensors can change, leading to physical measurable changes that reflect changes in the magnetic or dielectric properties of the surroundings [3].

Split-ring resonators (SRRs) and complementary split-ring resonators, which are sub-wavelength resonators, have been utilized to develop various sensing modalities [4,5]. Examples include biochemical applications [6,7,8,9], anomaly detection in metals and dielectric materials [10,11,12], microfluidic applications [13,14,15], dielectric characterization [6,16,17,18,19,20,21,22,23], and fluid concentration detection [24]. However, planar sub-wavelength resonators have a limitation in their effective sensing regions, which affects the sensor’s sensitivity. This is because the material under test (MUT) does not interact effectively with the highest field concentration [25], and so the shift in resonance frequencies and overall sensitivity of the sensor are minimized. To overcome this limitation, sharp tips can be introduced and symmetry can be broken in the resonators (e.g., split-ring resonators) to increase the interaction with MUT [26]. Despite the fact that most of the energy is stored in the substrate, in [27], a significant enhancement in the resonators’ sensitivity can be achieved by creating a substrate channel that accommodates MUT and enables it to interact more effectively with the resonators. But, designing a channel in the sensitive volume can be challenging, and optimization may be needed to locate the optimal sensitive position of the channel inside the substrate [27]. Thus, it is essential to explore and investigate other techniques for sensitivity enhancement.

The feasibility of using the three-dimensional capacitor-based technique for sensitivity enhancement was reported in [28]. This technique can be combined with other resonators that are suitable for specific applications, especially those requiring smaller sensing regions. For instance, in [29], this technique was employed to detect fluids using split-ring resonators (SRRs). Complementary structures such as CSRRs have higher sensitivity [16]; however, they have larger effective sensing regions. Therefore, a new design that can effectively combine the advantages of smaller sensing regions and higher sensitivity is necessary to improve the overall performance of these resonators.

In this work, we present a new sensor design for achieving maximum sensitivity without requiring the material under test to interact with the entire sensing volume of resonators. The proposed design utilizes a trapezoid-shaped resonator (TSR) with metallic bars (which is effectively a three-dimensional capacitor) to enhance the electric field in the sensing region and increase the overall sensitivity. The trapezoid shape allows for the sensing region to be reduced while maintaining high sensitivity, while the metallic bars enhance the electric field in the sensing region. The TSR design combines the advantages of both split-ring resonators (SRRs) and complementary structures such as complementary electric-LC resonators (CELCRs). The geometry of the trapezoid shape has two bases that can be chosen to create smaller sensing regions, similar to SRRs, or even smaller yet with higher sensitivity. Compared to SRRs, TSRs can be designed in the ground plane of microstrip lines, making them advantageous for microfluidic applications where there will be no direct interaction with the strip lines if a micro-channel is placed in the sensing region. Thus, such designs can be adopted for microfluidic applications [3,30,31,32].

## 2. Trapezoid Shape and Three-Dimensional Capacitor for Sensitivity Enhancement

Complement structure-based sensors, such as CSRRs, have shown higher sensitivity [16,33], which can be associated with their larger physical sensing regions compared to SRRs. Indeed, SRRs have been adopted for applications that require smaller sensing regions such as microfluidic applications [3,30,31,32]. However, other complement structures, with more control in their topological structures, can be considered, such as CELCRs, which can be designed using a trapezoid shape to create smaller sensing regions. A trapezoid shape has two bases that can be chosen, so using two trapezoid shapes, one can design a complement structure-based resonator with a smaller sensing region. For understanding the benefit of trapezoid shapes, we will start by analyzing rectangular shapes.

Figure 1 shows a CELCR composed of two rectangular shapes, denoted as $R_{1}$ and $R_{2}$ [28]. The area that is critical for sensing is a × (*L*− 2a), denoted as S in Figure 1. To enable a high-sensitivity response, the sensing region must be fully covered with MUTs. However, for technologies that require smaller sensing regions or MUTs’ volumes on a micro-scale, CELCRs in the current topological form will not be suitable for such applications. For example, in microfluidic-based technology, the localization and selectivity of electromagnetic energy to a small region are essential if utilized for heating and sensing individual droplets [31,32,34].

Compared with the traditional CELCR shape, the trapezoidal shape can address the challenge of achieving smaller and more sensitive sensing areas. Thus, the trapezoidal resonator design combines the advantages of SRR and CELCR in terms of having smaller sensing regions and being highly sensitive. A micro-scale-based region can be achieved using two-trapezoid shapes denoted as $T_{1}$ and $T_{2}$, as shown in Figure 2a. In fact, at the limit where one of the bases of a trapezoid shape is approaching zero, the shape will become triangular; hence, a smaller region can be further achieved. The trapezoid-shaped resonator illustrated in Figure 2a is designed in the ground plane of a microstrip line through an etching process. When operating at the resonance frequency, there is a significant increase in the amount of electric energy stored in the proximity of the TSR [2]. The electric field in the substrate and the surrounding air can be modeled using an effective capacitance $C_{\mathrm{sub}}$ (representing the capacitance due to the electric fields in the substrate) and $C_{\mathrm{air}}$ (representing the capacitance due to the electric fields in the air) (see Figure 2b). Therefore, the total effective capacitance of the TSR can be obtained by adding these two capacitances together. If the resonator effective capacitance is denoted as $C_{R}$, then $C_{R}$ = $\alpha_{1}$$C_{\mathrm{sub}}$ + $\alpha_{2}$$C_{\mathrm{air}}$, where $\alpha_{1}$ and $\alpha_{2}$ are real numbers accounting for the contribution factor of $C_{\mathrm{sub}}$ and $C_{\mathrm{air}}$ to the total $C_{R}$; hence, $\alpha_{1}$ + $\alpha_{2}$ = 1.

Since the CELC resonator is a sub-wavelength resonator, it can be modeled and analyzed using lumped circuit elements. However, the model is only accurate near the resonance frequency. Figure 3 shows the diagram of a microstrip line used to excite the TSR. Figure 4 shows a circuit model for the proposed TSR. $L_{\mathrm{TL}}$ and $C_{\mathrm{TL}}$ represent the equivalent inductance and the capacitance per unit length of the transmission line (TL); whereas $C_{R}$, $R_{R}$, and $L_{R}$ represent the equivalent capacitance, resistance, and inductance of the resonator, respectively. Since the circuit model is similar to the circuit reported in [35], the resonance frequency is given as
(1)$$f_{z}=\frac{1}{2\pi \sqrt{L_{R}\left(C_{R}+C_{\mathrm{TL}}\right)}}$$ In terms of $C_{\mathrm{sub}}$ and $C_{\mathrm{air}}$, the resonance frequency can be expressed as
(2)$$f_{z}=\frac{1}{2\pi \sqrt{L_{R}\left(\alpha_{1}C_{\mathrm{sub}}+\alpha_{2}C_{\mathrm{air}}+C_{\mathrm{TL}}\right)}}=\frac{1}{2\pi \sqrt{L_{R}\left(\left(1-\alpha_{2}\right)C_{\mathrm{sub}}+\alpha_{2}C_{\mathrm{air}}+C_{\mathrm{TL}}\right)}}$$ Thus, if $\alpha_{2}$ is approaching 1, the contribution of $C_{\mathrm{air}}$ will be increased; hence, MUTs will be interrogated effectively with the electric field, leading to higher sensitivity.

Since the electric field will be focused in the material with the highest dielectric constant (the substrate), the total capacitance is largely determined by the effective capacitance $C_{\mathrm{sub}}$, as noted in [25]. Thus, it is expected that $\alpha_{1}$ will be higher. By placing MUT on the opposite side of the substrate in free space, the interaction with the resonators is minimized, resulting in a minimal shift in the resonance frequency and therefore minimizing the overall sensitivity of the sensor. To overcome this limitation, one can adopt the three-dimensional capacitor reported in [28,29]. Figure 5 shows the diagram of the sensor where a 3D capacitor is included by adding metallic bars that are extended into the free space and vertically from the sensor plane while being attached to the resonator. The metallic bars, denoted by their physical length as $L_{P}$, can be modeled as an additional capacitor ($C_{\mathrm{PP}}$). Figure 6 illustrates the corresponding circuit model updated with the additional element of $C_{\mathrm{PP}}$. $C_{\mathrm{PP}}$ is now a function of $L_{P}$, consequently, as $C_{\mathrm{PP}}$ will be added in parallel to $C_{R0}$, the resonance frequency, at the transmission zero, will be a function of $L_{P}$, given as
(3)$$f_{z}\left(L_{P},\epsilon_{\mathrm{MUT}},g,d\right)=\frac{1}{2\pi \sqrt{L_{R}\left(C_{R0}+C_{\mathrm{TL}}+C_{\mathrm{PP}}\left(L_{P},\epsilon_{\mathrm{MUT}},g,d\right)\right)}}$$
where $C_{R0}$ is the original capacitor of the resonator without the metallic bars, and $\epsilon_{\mathrm{MUT}}$ is the permittivity of MUT between the parallel bars. Note that $C_{R}$ = $C_{R0}$ + $C_{\mathrm{pp}}$. Therefore, the metallic bars will increase the contribution of $C_{\mathrm{air}}$ to the total capacitance of the resonator, $C_{R}$. Hence, the interaction is increased with MUT, which is expected to increase the sensitivity. Furthermore, based on the circuit model shown in Figure 6, the proposed resonator can be classified as a parallel resonator. The quality factor can be expressed as [14]
(4)$$Q=R_{R}\sqrt{\frac{C_{R0}+C_{\mathrm{TL}}+C_{\mathrm{PP}}}{L_{R}}}$$

From (Equation 4), it can be observed that $C_{\mathrm{PP}}$ will enhance the quality factor. However, since $C_{\mathrm{PP}}$ is a function of $\epsilon_{\mathrm{MUT}}$, which will be added in parallel to $C_{R0}$ and $C_{\mathrm{TL}}$, its equivalent losses will lower the overall losses, and consequently, lower the quality factor. In addition, by loading $C_{\mathrm{PP}}$ with MUT, the electric field will be more focused in MUT than the substrate, which will cause the lowering of the coupling (smaller $C_{\mathrm{TL}}$) between TL and the resonator. Thus, $C_{\mathrm{TL}}$ depends on $C_{\mathrm{PP}}$. Consequently, from (Equation 4), one can expect a trade-off between a higher quality factor and a higher coupling factor (represented in higher $C_{\mathrm{TL}}$).

## 3. Sensor Design, Numerical Analysis, and Discussion

Sub-wavelength resonators such as the trapezoid-shaped resonator can be excited using a quasi-TEM mode that can be generated using a microstrip line. The resonator is designed in the ground plane by etching out the topology depicted in Figure 2. For the purpose of comparisons with the resonator in [29], the dimensions are a = g = 0.5 mm, *L* = 7.5 mm, S = a × a ($a^{2}$ = 0.25 m$m^{2}$).

Since the sensor response will be evaluated using scattering parameters ($|S_{21}|$), which will be recorded using a vector network analyzer (VNA) with a 50 Ω input impedance, a 50 Ω microstrip line with a strip width of 1.63 mm was designed on a Rogers substrate (RO4350) with a thickness of $W_{\mathrm{TL}}$ = 0.76 mm, a loss tangent of 0.004, and a relative permittivity of 3.66. The dimensions that are shown in Figure 2 are $L_{\mathrm{TL}}$ = 100 mm and $W_{G}$ = 50 mm. The bars’ length ($L_{p}$) will be studied extensively by exploring different values to investigate its effects on the resonance frequency. The metallic bars are aluminum. Table 1 presents a summary of the design specification used in this work.

Figure 7a shows the $|S_{21}|$ of the sensor without the bars, over the frequency range of 4.5 to 7.5 GHz. The frequency at which $|S_{21}|$ is minimum is the resonance frequency, which has a value of 6.28 GHz (the resonance frequency was chosen for its suitability to our measurement setup and to facilitate comparison with the sensor reported in reference [29]). The calculated quality factor of the resonator without the bars is found to be approximately 43. Since the circuit model shown in Figure 4 is used to predict the resonance frequency, it is important to validate such a circuit. The procedure is as follows: The response of the sensor ($|S_{21}|$ and $|S_{11}|$) in the form of a touchstone file was extracted from the full-wave numerical simulation performed using ANSYS HFSS [36]. Then, the response was imported to the circuit simulator Keysight-ADS [37]. By using the optimization toolbox provided in ADS, the circuit elements shown in Figure 4 were extracted. Table 2 shows the extracted elements.

With the metallic bars, the resonance frequency becomes a function of the length of bars ($L_{P}$), as well as the gap between the bars (denoted as g in Figure 5). To investigate the effects of the bars’ length on the resonance frequency and the quality factor, $L_{P}$ was varied from 5 to 65 mm in increments of 5 mm. Since there will be an overlap between the response curves at different values of $L_{P}$, some values were selected for the plot as shown in Figure 8. However, one can plot the response ($|S_{21}|$) as a 2D image as shown in Figure 9. The image color ranges from deep yellow to deep blue, where deep yellow represents a full transmission, while deep blue represents a transmission zero. On the y-axes, the distance (in frequency) between the deep blue colors represents the changes in the resonance frequencies (the frequencies at transmission zero) due to the changes in the bars’ length ($L_{P}$). An example of $\Delta f_{z}$ is shown in Figure 9.

Plotting $|S_{21}|$ as an image will help to observe the width of the minimum transmission (in frequency) at which the quality factor of the resonator can be observed. As was expected from (Equation 4), and quantitatively proven using the numerical simulation, it is evident that the bars contribute to the enhancement of the quality factor. For the ranges of $L_{P}$ from 5 to 65 mm, the quality factor is given in Figure 10.

The circuit model shown in Figure 6 was validated using the same previous procedure. Note that the chosen $L_{P}$ was 10 mm (other values can be alternatively chosen). The extracted circuit elements are presented in Table 2. The response of the circuit model versus the numerical simulation is shown in Figure 11. From Table 2, one can observe that the values of $L_{R}$ and $C_{R}$ are increased by 59% and 167%, respectively. The increment in both values caused the resonance frequencies to shift down to lower frequencies. In the case of $L_{P}$ = 10, the resonance frequency was shifted from 6.28 GHz (no bars) to 3.51 GHz, which is approximately 44%.

It is expected that the total electric field will be enhanced between the bars. To quantify this enhancement, the field was calculated using HFSS. The electric field was calculated along the virtual line shown in Figure 12a for $L_{P}$ = 30 (which corresponds to the highest quality factor). The total calculated electric field of the TSR with and without the metallic bars is shown in Figure 13. It can be easily observed that the metallic bars concentrate the electric field in the sensing region. Therefore, as predicted by (Equation 3), the material between the metallic bars will strongly affect the resonance frequency.

The detection mechanism of the sensing system is through observing the changes in the resonance frequency due to the presence of MUT in the sensing region. To compare the sensitivity of the proposed sensor with other planar sensors such as the one recently reported in [29], the following equations were used,
(5)$$S=\frac{\Delta f_{z}}{f_{z0}\left(\epsilon_{r}-1\right)}\times 100$$
(6)$$\mathrm{Norm}\;\Delta f\;=\;\frac{\Delta f_{z}}{f_{z0}}\times 100$$
where $\Delta f_{z}$ represents the relative change in the resonance frequency compared to the reference case (i.e., air), $f_{z0}$ is the resonance frequency of the reference case, $\epsilon_{r}-1$ is the relative difference in the dielectric constant of the materials being studied and Norm Δf is the normalized resonance frequency shift.

To simulate a real-world application, the arms were immersed completely inside a cylindrical dielectric material with its permittivity being varied from 1 to 10 in increments of 0.5. Figure 14a,b show the resonance frequency shift ($\Delta f_{z}$) and the sensitivity of the sensor with and without the metallic bars versus the relative permittivity. Table 3 gives a performance comparison for the sensor with and without the bars.

The sensor was examined to detect the dielectric constant when the bars are immersed at different levels from 1 to 10 mm with an increment of 1 mm. Simultaneously, the dielectric constant was varied from 1 to 100 with an increment of 1 for the range 1 to 11, and an increment of 10 for the range 11 to 100. Figure 15 shows the resonance frequency shift ($\Delta f_{z}$) versus the relative permittivity at different immersion levels.

The proposed sensor was compared with the sensor reported in [29], where both resonators have identical bars’ lengths. Equations (Equation 5) and (Equation 6) will be used for the evaluation and comparison. Figure 16a,b show the resonance frequency shift and sensitivity of SRR and TSR with metallic bars of length 4 mm. Note that the dimensions of the SRR and the bars’ length were taken from [29]. By comparing the sensitivity of SRR and TSR, the enhancement in the sensitivity was quantified and presented in Figure 16b. It is evident that the TSR combines the advantages of SRR and CELCR of having smaller sensing regions and being highly sensitive. Table 4 provides a comprehensive comparison between different microwave sub-wavelength microwave sensors. Furthermore, from Figure 16a considering the case of TSR, there are three regions where $\Delta f_{r}$ as a function of $\epsilon_{r}$ behaves differently with the increment of $\epsilon_{r}$, where ${\mathrm{region}}_{3}$ indicates the approaching to a saturation region. To quantify the minimum detectable range, we utilized a fitting function technique to mathematically model $\Delta f_{r}$ as a function of $\epsilon_{r}$, which can be expressed as,
(7)$$\Delta f_{r}\left(\epsilon_{r}\right)=3.21541-1.7812e^{\left(\frac{-\epsilon_{r}}{1.19663}\right)}-2.8787e^{\left(\frac{-\epsilon_{r}}{6.05237}\right)}$$ The change in $\Delta f_{r}$ due to a small change in $\epsilon_{r}$ can be expressed as,
(8)$$\frac{d\Delta f_{r}\left(\epsilon_{r}\right)}{d\epsilon_{r}}=1.48851e^{\left(\frac{-2\epsilon_{r}}{1.19663}\right)}+0.4756e^{\left(\frac{-2\epsilon_{r}}{6.05237}\right)}$$ Thus, one can use Equation (Equation 8) such that
(9)$$min\left\{\frac{d\Delta f_{r}\left(\epsilon_{r}\right)}{d\epsilon_{r}}\right\}=min\left\{1.48851e^{\left(\frac{-2\epsilon_{r}}{1.19663}\right)}+0.4756e^{\left(\frac{-2\epsilon_{r}}{6.05237}\right)}\right\}$$
and solve it numerically to find $\epsilon_{r}$ at which $\Delta f_{r}$ is minimum. Since ${\mathrm{region}}_{3}$ is approaching saturation, it is expected that the minimum detectable values will depend on the resolution of the apparatus, e.g., VNA, which is used to measure the response of the system. At the limit where the value of Equation (Equation 9) is zero, the system cannot distinguish between two close values of $\epsilon_{r}$.

## 4. Fluid-Level Measurements

This part focuses on the fabrication and testing of TSR sensors for fluid detection and fluid-level measurement. The sensors were fabricated using printed circuit board (PCB) technology. The design specifications of the sensors are presented in Table 1. To enhance the sensitivity, metallic bars with a length of ($L_{P}$) = 30 mm were utilized as this parameter was found to exhibit the highest quality factor. While other metallic materials with higher conductivity, such as gold and silver, could be employed, aluminum material was chosen for these bars due to its low cost and ease of fabrication. Since it is difficult to solder aluminum to copper, a conductive glue from the manufacturer MG Chemicals (8331-14G) was used. The glue is a 2-part epoxy with a 1:1 mix ratio, a working time of 10 min, and a resistivity of 7.0 × $10^{3}$Ω·cm. It is recommended by the manufacturer to allow 24 h to cure at room temperature or to use an oven by following one of these options (time/temperature): 15 min at 65 Celsius or 7 min at 125 Celsius. The glue was used to attach the bars to the surface plane of the sensor. Note that from (Equation 3), it is evident that a smaller value of the distance between the bars (g) will lead to a higher capacitance of $C_{\mathrm{PP}}$, which will shift the resonance frequency to a lower frequency. With the use of the conductive glue, one can tune the resonance frequency to an intended one by optimizing the distance between the bars, g. To assess the enhanced sensitivity, the two resonators, with and without the aluminum bars, were tested. Figure 17 shows the fabricated TSR sensor.

The procedure for the fluid detection involved immersing the sensor bars inside two different fluids, chloroform and dichloromethane, with permittivities of 4.81 and 8.93, respectively (taken from [42]). We also tested other materials with higher dielectric constants, such as distilled water, to evaluate the sensor’s performance under different conditions. The immersion levels of the bars were controlled using an XYZ-positioning stage with a resolution of 0.01 mm in the Z-direction, as shown in Figure 17d. The data were collected using a 50 Ω vector network analyzer. The network analyzer was calibrated first using a short-open-load-thru calibration kit to ensure accurate measurements. The collected data were then analyzed to determine the sensor’s sensitivity and accuracy in measuring fluid levels. The response of the sensor with the bars placed in the air is shown in Figure 18. The results show a strong agreement between the simulation and the measurements.

To quantify the enhanced sensitivity of the sensor, first, the TSR sensor was tested without the bars (shown in Figure 17c). The plastic container (encompassing the sensing region) was filled with the fluids under test. The results of this test are given in Figure 19 showing a sensitivity of 0.92% for chloroform and 0.66 % for dichloromethane. The experiment was repeated with the TSR sensor with the bars using the same fluids and glass beaker, as shown in Figure 12b. The bars were completely immersed in the fluids. The result of this test is presented in Figure 20, showing a sensitivity of 14.1% for chloroform and 8.3% for dichloromethane. Therefore, the enhancement in sensitivity of the TSR sensor is 1435% for chloroform and 1168% for dichloromethane.

For the fluid-level measurement, the bars were gradually immersed inside a glass beaker containing chloroform, with an immersion increment of 0.76 mm (note that the first step was 0.5 mm). The resonance frequency at each step versus the step values was used to create a calibration curve, as shown in Figure 21. Calibration curves were also generated for dichloromethane and distilled water using the same procedure, as illustrated in Figure 22.

## 5. Conclusions

In this work, we proposed a new design for planar sub-wavelength resonators that combines the advantages of split ring resonators (SRR) and complementary electric-LC resonators (CELCR) by incorporating metallic bars in a trapezoid-shaped resonator. The trapezoid shape allows for a smaller sensing region, while the metallic bars enhance the electric field in the sensing region, resulting in higher sensitivity. Numerical simulations were used to design and evaluate the sensor. The sensor was fabricated and validated using PCB technology with aluminum bars. The testing of the sensor’s sensitivity demonstrated that it can be used for microfluidic and level detection applications, with the enhanced sensitivity achieved through the use of metallic bars. The calibration curves generated for fluid-level measurement can serve as a basis for future applications in fluid-level detection and related fields. One of the main advantages of this new design is that it offers higher sensitivity than previous designs while also having a smaller sensing region, making it well-suited for microfluidic applications. Additionally, the use of aluminum bars makes the sensor more affordable and easier to fabricate. 

## Figures and Tables

**Figure 1 sensors-23-06453-f001:**
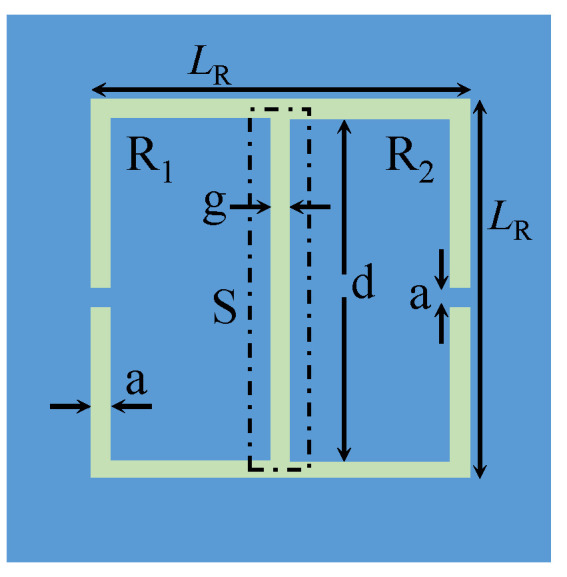
CELCR designed using two-rectangular shapes denoted as $R_{1}$ and $R_{2}$.

**Figure 2 sensors-23-06453-f002:**
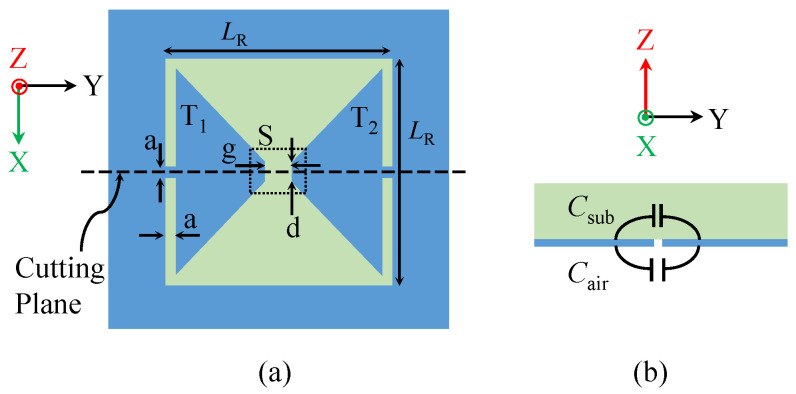
(**a**) TSR designed using trapezoid shapes denoted as $T_{1}$ and $T_{2}$. (**b**) Side view of TSR through the cutting plane shown in Figure 2a.

**Figure 3 sensors-23-06453-f003:**
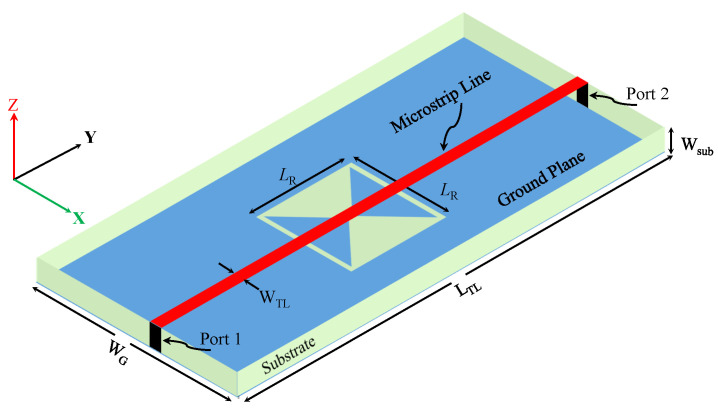
A perspective view of the TSR sensor geometry.

**Figure 4 sensors-23-06453-f004:**
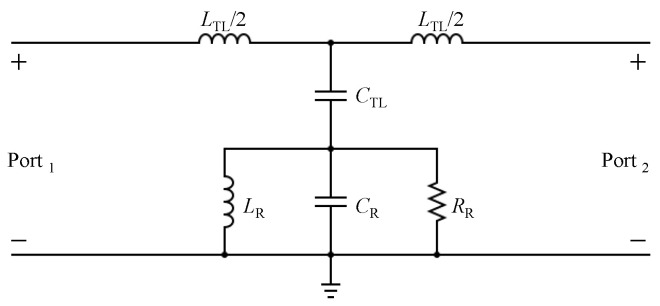
Two-port circuit model of the microstrip line used to excite the TSR, where $L_{\mathrm{TL}}$ and $C_{\mathrm{TL}}$ are the equivalent inductance and the capacitance per unit length of the transmission line (TL), and $C_{R}$, $R_{R}$, and $L_{R}$ are the equivalent capacitance, resistance, and inductance of the resonator, respectively.

**Figure 5 sensors-23-06453-f005:**
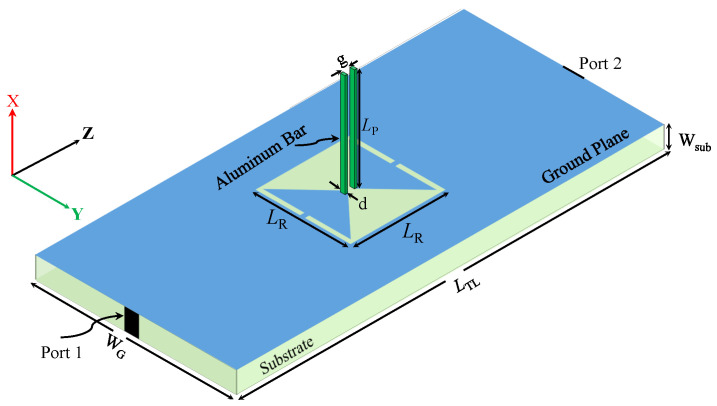
Perspective view of a TSR sensor loaded with aluminum bars.

**Figure 6 sensors-23-06453-f006:**
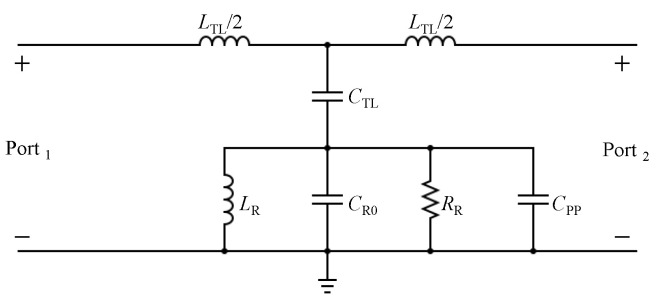
Circuit lumped-element model of a transmission line (TL) used to excite TSR, where $L_{\mathrm{TL}}$ and $C_{\mathrm{TL}}$ are the equivalent capacitance and the inductance of TL, $L_{R}$, $C_{R}$ ($C_{R}$ = $C_{R0}$ + $C_{\mathrm{pp}}$), and $R_{R}$ are the equivalent inductance, capacitance, and resistance of the resonator, respectively. $C_{\mathrm{PP}}$ is the equivalent capacitance of the metallic bars.

**Figure 7 sensors-23-06453-f007:**
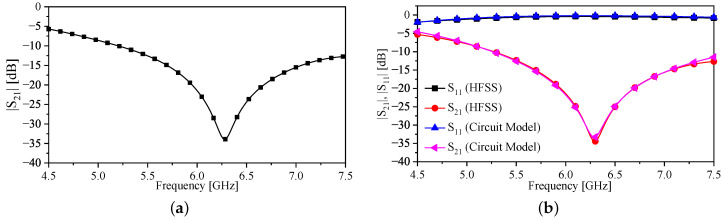
(**a**) The response ($|S_{21}|$) of the TSR sensor without the metallic bars extracted using numerical simulation. (**b**) A comparison of the response obtained from the numerical simulation and the circuit model.

**Figure 8 sensors-23-06453-f008:**
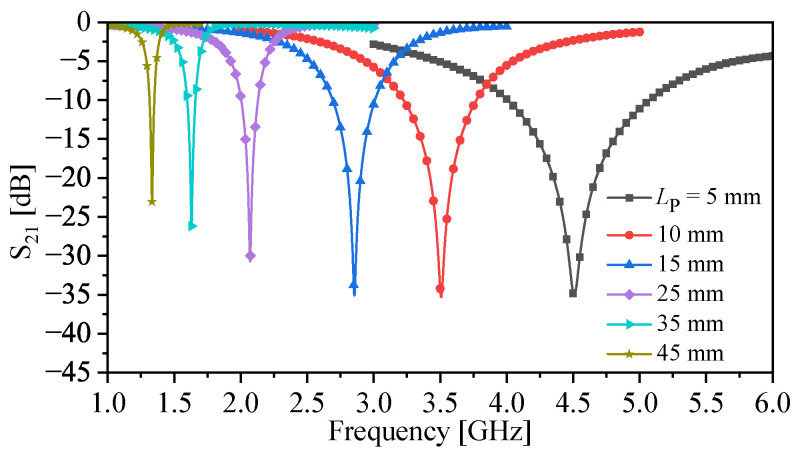
Response ($|S_{21}|$) of the sensors at different values of $L_{P}$ = 5, 10, 15, 25, 35, and 45 mm.

**Figure 9 sensors-23-06453-f009:**
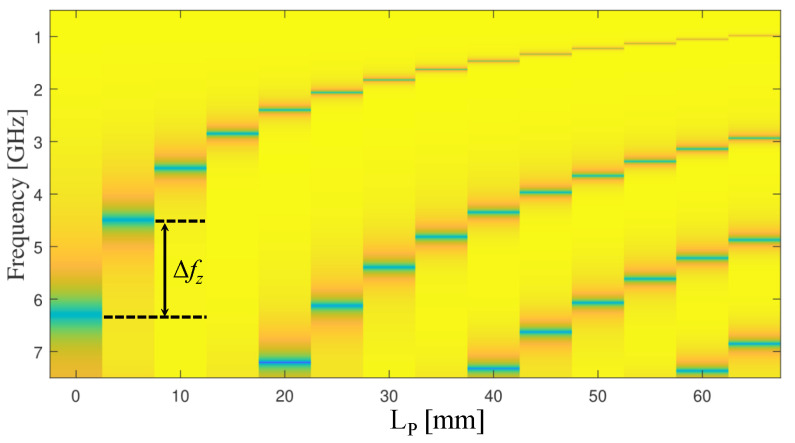
2D image of the response ($|S_{21}|$) at different values of $L_{P}$ = 0, 5, 10, 15, 25, 35, and 45 mm.

**Figure 10 sensors-23-06453-f010:**
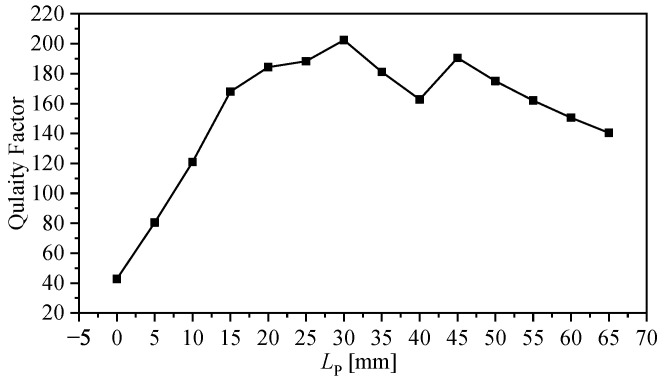
Quality factor as a function of the bars’ length, $L_{P}$.

**Figure 11 sensors-23-06453-f011:**
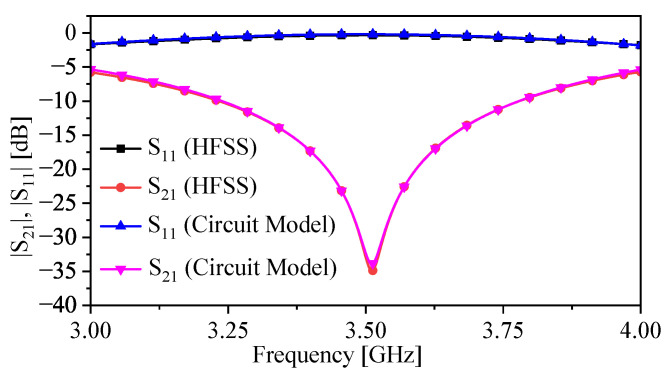
Response of the circuit model and the numerical simulation with the bar’s length of $L_{P}$ = 10.

**Figure 12 sensors-23-06453-f012:**
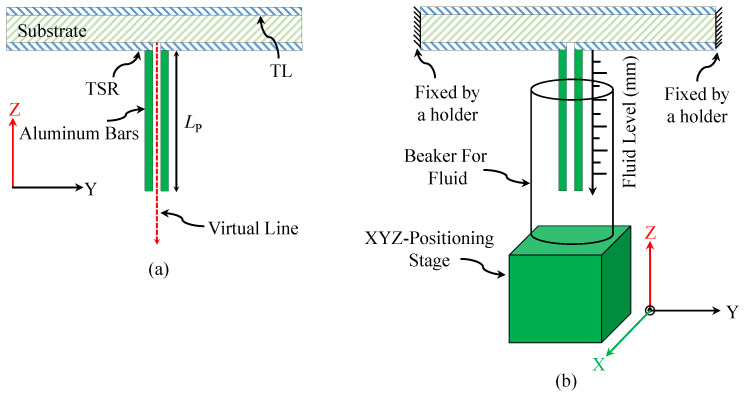
Side view of the sensor presenting (**a**) the virtual line used to plot the total electric field between the bars (**b**) the schematic of the experimental setup used for the measurement.

**Figure 13 sensors-23-06453-f013:**
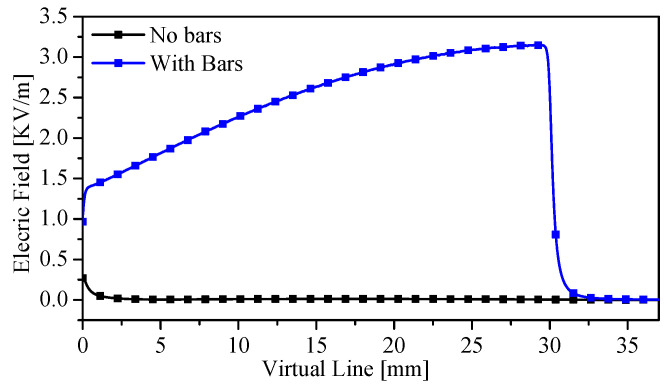
Total electric field between the bars along the virtual line shown in Figure 12a.

**Figure 14 sensors-23-06453-f014:**
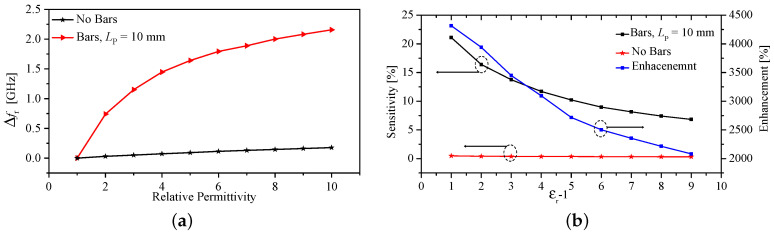
(**a**) Resonance frequency shift ($\Delta f_{z}$). (**b**) The sensitivity, and the enhancement in the sensitivity of the proposed sensor with and without the metallic bars.

**Figure 15 sensors-23-06453-f015:**
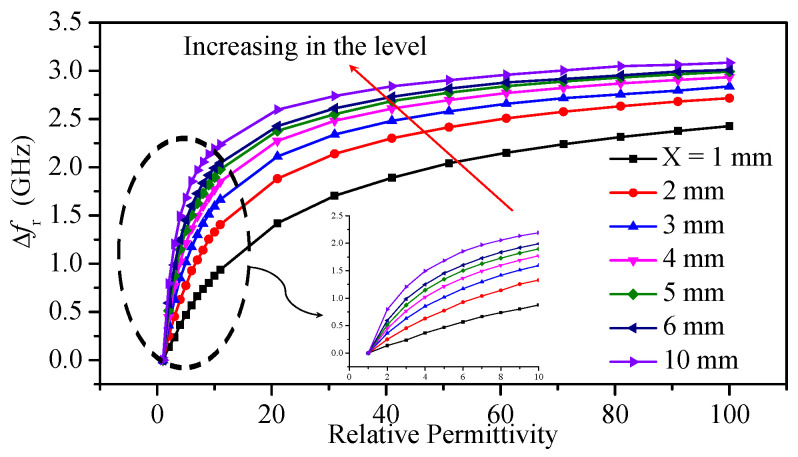
Resonance frequency shift as a function of the relative permittivity for different immersion levels and at different dielectric constants.

**Figure 16 sensors-23-06453-f016:**
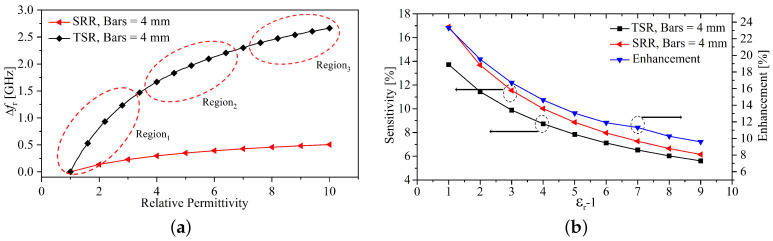
(**a**) Resonance frequency shift ($\Delta f_{z}$). (**b**) Sensitivity, and the enhancement in the sensitivity of the proposed sensor with and without metallic bars.

**Figure 17 sensors-23-06453-f017:**
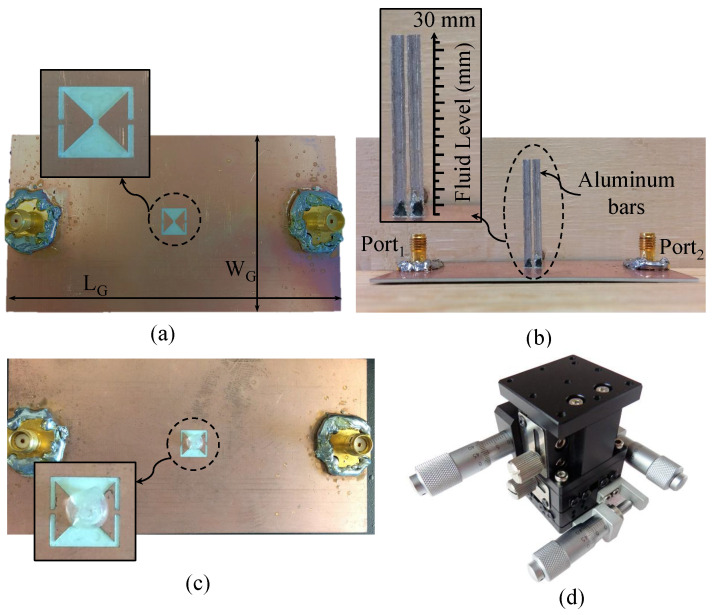
The fabricated TSR sensors. (**a**) Without the bars. (**b**) With the bars. (**c**) The sensor without the bars in a plastic fluid container. (**d**) Manual-XYZ-positioning stage.

**Figure 18 sensors-23-06453-f018:**
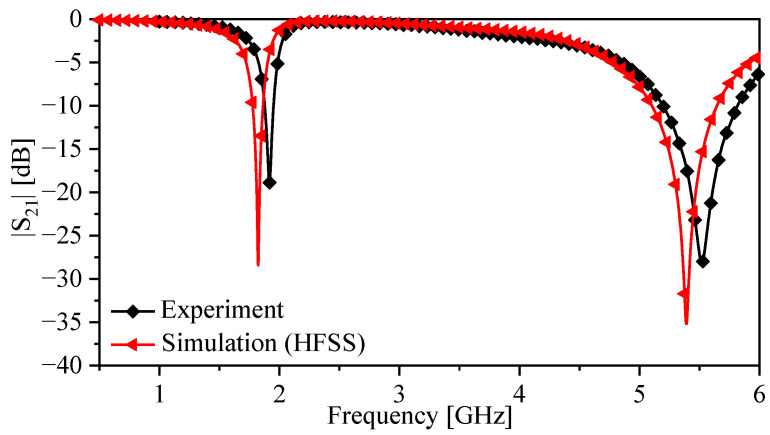
Measured $|S_{21}|$ over the frequency range 0.3 to 6 GHz for the TSR sensor with bars.

**Figure 19 sensors-23-06453-f019:**
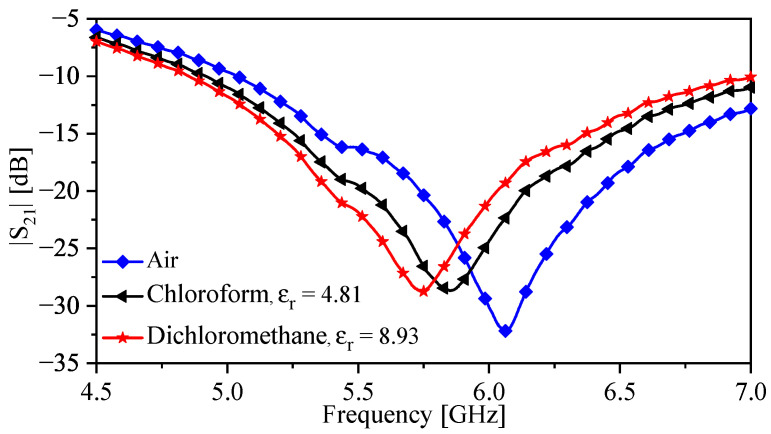
Measured $|S_{21}|$ for the sensor without the bars, in the presence of air, chloroform, and dichloromethane.

**Figure 20 sensors-23-06453-f020:**
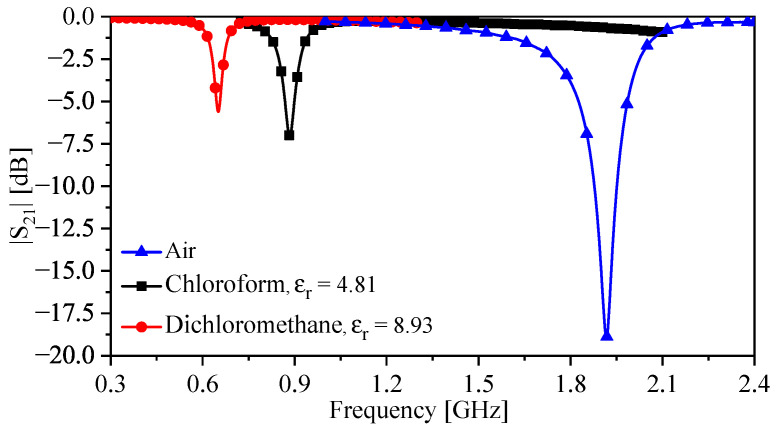
Measured $|S_{21}|$ for the sensor with the bars, in the presence of air, chloroform, and dichloromethane.

**Figure 21 sensors-23-06453-f021:**
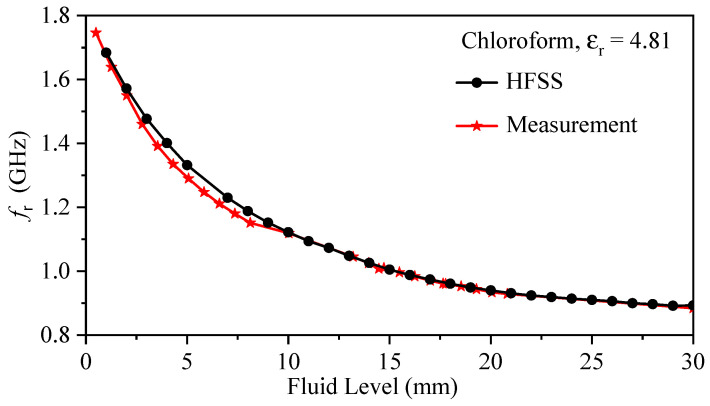
Calibration curve generated by the experiment and simulation for measurement of chloroform.

**Figure 22 sensors-23-06453-f022:**
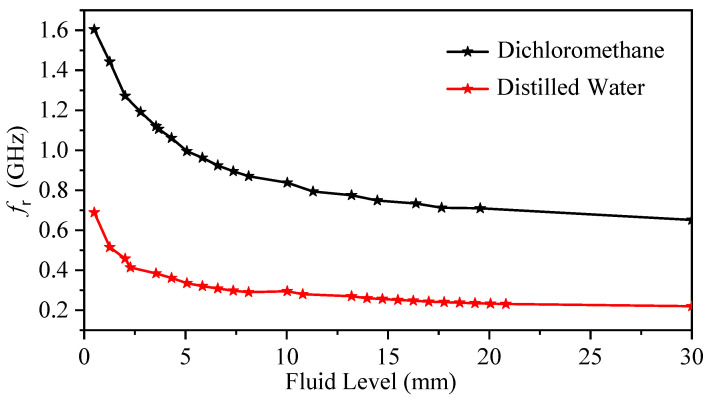
Calibration curves for the dichloromethane and distilled water generated by the experiment.

**Table 1 sensors-23-06453-t001:** Design Criteria.

Sensor Type	a (mm)	$L_{R}$ (mm)	$W_{\mathrm{TL}}$ (mm)	$W_{G}$ (mm)	$L_{\mathrm{TL}}$ (mm)	$W_{\mathrm{sub}}$ (mm)	$L_{P}$ (mm)	g (mm)	d (mm)
**CELCR (No Bars)**	0.5	7.5	1.63	50	100	0.762	NA	0.5	0.5
**TSR (No Bars)**	0.5	7.5	1.63	50	100	0.762	NA	0.5	0.5
**TSR (with Bars)**	∼	∼	∼	∼	∼	∼	Vari.	0.5	0.5

**Table 2 sensors-23-06453-t002:** Extracted circuit elements.

Sensor Type	$L_{\mathrm{TL}}$ (nH)	$C_{\mathrm{TL}}$ (pF)	$L_{R}$ (nH)	$C_{R}$ (pF)	$R_{R}$ (kΩ)
**The trapezoid-shaped resonator (No Bars)**	1.176	1.288	0.252	1.25	0.376
**The trapezoid-shaped resonator (with Bars)**	1.332	1.782	0.401	3.341	0.934

**Table 3 sensors-23-06453-t003:** Sensitivity comparison between the sensor with and without metallic bars.

$\epsilon_{r}-1$	Trapezoid-Shaped Resonator (No Bars)	Trapezoid-Shaped Resonator (with Bars)
1	0.477%	21.094%
2	0.406%	16.405%
3	0.387%	13.749%
4	0.366%	11.701%
5	0.363%	10.228%
6	0.345%	8.979%
7	0.332%	8.152%
8	0.320%	7.418%
9	0.313%	6.835%

**Table 4 sensors-23-06453-t004:** A comprehensive comparison between microwave sub-wavelength microwave sensors.

Ref.	$f_{r}$ [GHz]	Tunable	F. Level	MAX. S [%]	L. Dielectric	H. Dielectric	Relative Size	Geometry
[38]	20	NO	No	0.347		Yes	0.086 $\lambda_{0}$	Planar
[39]	1.35	NO	No		Yes	Yes	0.25 $\lambda_{0}$	Planar
[40]	3.1	NO	No	0.187	Yes	Yes	0.65 $\lambda_{0}$	Planar
[41]	1.91	NO	No			Yes	0.06 $\lambda_{0}$	Planar
[13]	2.1	NO	No			Yes	0.049 $\lambda_{0}$	Planar
[29]	2.828	NO	Yes	13.719	Yes	No	0.0707 $\lambda_{0}$	Planar with metallic bars
**[T.W]**	**4.812**	**NO**	**Yes**	**21.094**	**Yes**	**Yes**	**0.12 $\lambda_{0}$**	**Planar with metallic bars**

## Data Availability

Data generated during the study are contained within the article.

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
