# Peer review of "A Highly Sensitive 3D Resonator Sensor for Fluid Measurement"

_sensors, 2023, doi:10.3390/s23146453_

Round 1

Reviewer 1 Report

Title is interesting however, research is not unique. Before acceptance authors need to clarified the following comments:

1. Based on the manuscript, I didn't see  any novelty except the design. Authors must be clarify about the novelty. Design can't be a novelty only.

2. Most of the references are last 10 years before, meaning that the following research is established. Authors should be carefully checked about this.

3. A lot of references comes from authors and co-authors works, I think better authors compared their work with the outsiders of the author.

4. There is no measurement picture, how to detect the sensing. Authors must be provided the illustrations of the measurement set up.

5. Design of the unit cell information is not clear. How authors come to this design for using the sensor?

6. Why the results in discrepency in Fig. 18? Authors must be provided the details criticism.

7. Authors already provided in Table 1 sensor design parameter, but were not provided the parameters of CELCR designed and TSR designed values.

Authors must be provided the design parameters of the unit cell of these?

8. Authors provided the equivalent circuit but reviewer suggested to draw the equivalent circuit with values and compare with the either simulation or measured results? This is one kind of validation.

9. How the developed sensor will be practically implemented? What is the practical application?

Minor Check Spell is required.

Author Response

Reviewer 1:

  • Title is interesting however, research is not unique.

Answer R1-1

  • We appreciate the reviewer's comment regarding the title of the paper. We would like to address the reviewer's concerns about the uniqueness and novelty of our research.
  • If the reviewer is questioning the originality of the work, we would like to clarify that our work is indeed original and unique. Our proposed sensor is based on a new design that has not been previously reported in the literature. In fact, we have compared our proposed sensor to our previous work in references [28, 29], to highlight the strength of our new design.
  • However, if the reviewer is referring to the fact that the same problem has been addressed by other works in the literature, we have already acknowledged this in the manuscript. We have provided a comprehensive literature review and have discussed the limitations of previous works in addressing the problem. We emphasize that our proposed approach is a significant improvement over previous methods, which makes it a valuable and a strong contribution to the field.

Reviewer 1:

  • Based on the manuscript, I didn't see any novelty except the design. Authors must be clarify about the novelty. Design can't be a novelty only.

Answer R1-2

  • We appreciate the reviewer's comment regarding the novelty of our work. We would like to clarify that our proposed sensor is indeed novel and unique, not just in terms of its design, but also in terms of its sensitivity. We emphasize that not all new/novel designs represent an advancement in the state of the art or the state of the technology, but the significance and real-world impact of the new design is what matters. Our design is not only a novel one, but provides a significant enhancement in sensitivity over previous works.
  • We acknowledge that the sensitivity of the sensor is a crucial aspect of its performance, and we have presented a comprehensive comparison between our proposed sensor and a recently published sensor in reference [29]. As mentioned in section 3, paragraph 11 of the manuscript, we have used Equations (5) and (6) to evaluate and compare the resonance frequency shift and sensitivity of our proposed sensor with that of the sensor reported in [29]. We have also provided a quantitative comparison of different microwave sub-wavelength sensors in Table 4.
  • We would like to draw the reviewer's attention to Figure 16a and b, which show the resonance frequency shift and sensitivity of our proposed sensor (TSR) and the sensor reported in [29] (SRR), with metallic bars of length 4 mm. By comparing the sensitivity of SRR and TSR, we have quantified the enhancement in the sensitivity of our proposed sensor.

Reviewer 1:

  • Most of the references are last 10 years before, meaning that the following research is established. Authors should be carefully checked about this.

Answer R1-3

  • We appreciate the reviewer's comment regarding the references cited in the manuscript. We would like to clarify that we have carefully selected the references that are most relevant to our work. We have added these references to provide context for our research, and to highlight the strengths and weaknesses of previous methods.
  • Regarding the reviewer's comment that "most of the references are last 10 years before," we would like to emphasize that the age of a reference is not necessarily an indicator of its relevance or importance. We have made a concerted effort to include all relevant works, regardless of their publication date.
  • If the reviewer has identified a specific reference or references that he/she believe should have been included in the manuscript, we would be grateful if he/she could provide this information.

 Reviewer 1:

  • Most of the references A lot of references comes from authors and co-authors works, I think better authors compared their work with the outsiders of the author.

Answer R1-4

  • We appreciate the reviewer's comment regarding the references cited in the manuscript. We would like to clarify that we have included all relevant works, regardless of whether they were authored by ourselves or by others. From a strict academic point of view, it is important to provide a comprehensive literature review that covers all relevant works, including those authored by the authors of the manuscript.
  • Regarding the reviewer's comment that "a lot of references come from authors and co-authors works," we would like to point out that only nine out of the 42 cited works are authored by the coauthors of the manuscript.
  • Nevertheless, we appreciate the reviewer's feedback and we agree that it is important to compare our work with that of other researchers outside our team. We have already done so in the manuscript, and have provided a comparison between our proposed sensor and a recently published sensor in reference [13,38,39,40,41], which was not authored by us. Again, if the reviewer has identified a specific reference or references that he/she believe should have been included in the manuscript, we would be grateful if he/she could provide this information.

Reviewer 1:

  • There is no measurement picture, how to detect the sensing. Authors must be provided the illustrations of the measurement set up.

Answer R1-5

  • We appreciate the reviewer's comment regarding the measurement setup in our manuscript. We have provided a detailed description of how the measurement was conducted, and have included a schematic of the procedure in Figure 12. Furthermore, we have provided a photograph of the sensor and of the XYZ-positioning stage that was used in the measurement, to give readers a better understanding of the experimental setup. We believe that these details provide sufficient information for readers to understand how the measurement was conducted and how the sensing was detected. Unfortunately, the experiments were carried out more than 2 years ago and is not feasible to replicate at this time.  However, we believe the paper contains all the details for anyone to fully replicate the experiment.

Reviewer 1:

  • Design of the unit cell information is not clear. How authors come to this design for using the sensor?

Answer R1-6

We appreciate the reviewer's comment regarding the design of the unit cell in our manuscript. We would like to clarify that we have already provided a detailed explanation of how we came up with the design of the cell. This can be found in section 2, paragraphs 2 and 3 of the original manuscript:

“Fig. 1 shows a CELCR composed of two rectangular shapes, denoted as R1 and R2 [28]. The area that is critical for sensing is a×(L-2a), denoted as S in Fig. 1. To enable a high- sensitivity response, the sensing region must be fully covered with MUTs. However, for technologies that require smaller sensing regions or MUTs’ volumes are on a micro-scale, CELCRs in the current topological form will not be suitable for such applications. For example, in microfluidic-based technology, the localization and selectivity of electromag- netic energy to a small region are essential if utilized for heating and sensing individual droplets [31,32,34].

A micro-scale-based region can be achieved using two-trapezoid shapes denoted as T1 and T2, as shown in Fig. 2a. In fact, at the limit where one of the bases of a trapezoid shape is approaching zero, the shape will become triangular, hence a smaller region can be further achieved. The trapezoid-shaped resonator illustrated in Fig. 2a is designed in the ground plane of a microstrip line through an etching process. When operating at the resonance frequency, there is a significant increase in the amount of electric energy stored in the proximity of the TSR [2]. The electric field in the substrate and the surrounding air can be modeled using an effective capacitance Csub (representing the capacitance due to the electric fields in the substrate) and Cair (representing the capacitance due to the electric fields in the air) (see Fig. 2b). Therefore, the total effective capacitance of the TSR can be obtained by adding these two capacitances together. If the resonator effective capacitance is denoted as CR, then CR = α1Csub + α2Cair, where α1 and α2 are real numbers accounting for the contribution factor of Csub and Cair to the total CR, hence, α1 + α2 = 1.”

 Reviewer 1:

  • Why the results in discrepency in Fig. 18? Authors must be provided the details criticism.

Answer R1-7

  • We appreciate the reviewer's comment regarding the discrepancy in the results shown in Figure 18 of our manuscript. We would like to clarify that this discrepancy is mainly due to the fabrication tolerance and the use of conductive glue to attach the aluminum bars. Despite this discrepancy, we believe that there is strong agreement between the measurement and numerical simulation results, taking into account these factors. Furthermore, it is important to note that the results of the experiment for fluid detection do not depend on the numerical results, since calibration curves are generated from the experimental results. These calibration curves are shown in Figures 21 and 22 of the manuscript.

Reviewer 1:

  • Authors already provided in Table 1 sensor design parameter, but were not provided the parameters of CELCR designed and TSR designed values. Authors must be provided the design parameters of the unit cell of these?

Answer R1-8

  • We appreciate the reviewer's comment regarding the design parameters of the unit cell in our manuscript. We would like to clarify that while we have provided some design parameters in Table 1, we acknowledge that some parameters were missing.
  • We have updated Table 1 to include the missing design parameters of the unit cell, including those for the CELCR and TSR designs. We believe that this information will be helpful for readers to better understand our research and the design of the unit cell.
  • Figure 1, 2 and 3 are also updated to include the parameters, g, d, and WTL.

Reviewer 1:

  • Authors provided the equivalent circuit but reviewer suggested to draw the equivalent circuit with values and compare with the either simulation or measured results? This is one kind of validation.

Answer R1-9

  • We appreciate the reviewer's comment regarding the equivalent circuit in our manuscript. We would like to clarify that we have provided circuit models in Figure 6, extracted parameters in Table 2, and numerical results from the circuit model and HFSS in Figures 7 and 11.
  • While we have not explicitly drawn the equivalent circuit with values and compared it with simulation or measured results, we believe that the information we provided is sufficient to validate our approach. The circuit models and extracted parameters were used to simulate and predict the behavior of the sensor, and the numerical results from the circuit model and HFSS were compared to validate the accuracy of our predictions.

Reviewer 1:

  • How the developed sensor will be practically implemented? What is the practical application?

Answer R1-10

  • We appreciate the reviewer's question regarding the practical implementation and application of our proposed sensor. We would like to clarify that the sensor can be used to detect fluid levels in industrial applications. In addition, it can be utilized in laboratory settings to detect fluid concentration.
  • Furthermore, the design of the sensor allows for the creation of smaller and more sensitive sensing areas, which makes it suitable for applications in microfluidics. In such applications, the size of the sensing area is critical for detecting or heating individual droplets.
  • We believe that our proposed sensor has potential applications in a wide range of fields, including but not limited to chemical and biological sensing, environmental monitoring, and industrial process control.

Reviewer 2 Report

Based on the concept of three-dimensional capacitors, the authors modeled and fabricated a trapezoid-shaped resonator with out-of-plane metallic bars with the purpose of confining the electric field, thus enhancing device sensitivity.

In [1], the authors demonstrated that metal bars can be used as an add-on to increase the sensitivity of split-ring resonators. Based on their findings, they hypothesized that the same concept could be applied to other types of resonators. In this manuscript, the authors extended their previous work and partially validated the hypothesis previously mentioned by following a similar set of protocols and experiments (as described in [1]).

The main contribution of this manuscript is the demonstration that metal bars can be used to enhance the performance of trapezoid-shaped resonators (in addition to split-ring resonators). However, in comparison with the authors’ previous work [1], there is a lack of scientific novelty.

Author Response

Reviewer 2:

  • Based on the concept of three-dimensional capacitors, the authors modeled and fabricated a trapezoid-shaped resonator with out-of-plane metallic bars with the purpose of confining the electric field, thus enhancing device sensitivity. In [1], the authors demonstrated that metal bars can be used as an add-on to increase the sensitivity of split-ring resonators. Based on their findings, they hypothesized that the same concept could be applied to other types of resonators. In this manuscript, the authors extended their previous work and partially validated the hypothesis previously mentioned by following a similar set of protocols and experiments (as described in [1]). The main contribution of this manuscript is the demonstration that metal bars can be used to enhance the performance of trapezoid-shaped resonators (in addition to split-ring resonators). However, in comparison with the authors’ previous work [1], there is a lack of scientific novelty.

Answer R2-1

  • We appreciate the reviewer's comments regarding the novelty of our current work compared to our previous work in [29]. While it is true that the concept of using metallic bars to enhance the sensitivity of resonators has been explored in previous research, we believe that our work provides a roadmap for designers to find appropriate sensing areas for their designs based on their requirements, such as sensitivity and compactness.
  • Furthermore, our work demonstrates that the same concept of using metallic bars can be applied to other types of resonators, such as trapezoid-shaped resonators, and shows the effectiveness of this approach in enhancing the performance of the sensor.
  • We also note that our work addresses the challenge of achieving smaller and more sensitive sensing areas, which is critical for applications in microfluidics where the size of the sensing area matters for detecting or heating individual droplets. Please, refer to references [31,32,34]. Our proposed trapezoid-shaped resonator design allows for the creation of smaller and more sensitive sensing areas while also ensuring that there is no interaction between the MUT and the transmission line.
  • Overall, we believe that our work provides a valuable contribution to the field of sensor design and offers important insights for future research in this area.

Reviewer 3 Report

The authors presented a new design for planar sub-wavelength resonators that combines the advantages of split ring resonators and complementary electric-LC resonators by incorporating metallic bars in a trapezoid-shaped resonator. The authors performed numerical simulation to optimize the performance or the resocator. Overall, the work is interesting. A few minor points are listed below:

1. The authors should explain more clearly while the trapezoidal design is better than the rectangle shape in traditional CELCR.  Maybe compare the new design with the traditional CELCR experimentally.

2. What's the minimum sensing range of the new design? The author should mention this.

3. More details about the fabrication of new sensing system should be given. For example, the integration of the Al bars.

4. Scale bars should be added in Fig. 17.

Round 2

Reviewer 1 Report

All comments are addressed.

Minor Check spell is required.